
# Introduction to EarthCARE synthetic data using a global storm-resolving simulation

Woosub Roh[1], Masaki Satoh[1], Tempei Hashino[2], Shuhei Matsugishi[1], Tomoe Nasuno[3], and Takuji Kubota[4]

[1]Atmosphere and Ocean Research Institute, The University of Tokyo, Kashiwa, Chiba 277-8564, Japan
[2]Kochi University of Technology, Kami, Japan
[3]Research Institute for Global Change, Japan Agency for Marine-Earth Science and Technology, Kanagawa, Japan
[4]Earth Observation Research Center, Japan Aerospace Exploration Agency, Tsukuba, Ibaraki, Japan

*Correspondence to*: Woosub Roh (ws-roh@aori.u-tokyo.ac.jp)

**Abstract.** Pre-launch simulated data to be obtained from new sensors on a satellite is useful to develop retrieval algorithms and aid the rapid release of retrieval products after launch. Here we introduce Japanese Aerospace Exploration Agencies (JAXA) EarthCARE synthetic data based on simulations using a 3.5 km horizontal-mesh global storm-resolving model. Global aerosol transport simulation results are added for aerosol retrieval developers. Synthetic data were produced for four types of EarthCARE sensor: a 94 GHz cloud-profiling radar (CPR), a 355 nm atmospheric lidar (ATLID), a seven-channel multispectral imager (MSI), and a broadband radiometer (BBR). JAXA EarthCARE synthetic data include a standard product with data for two orbits and a research product with shorter frames and more detailed instrument settings. In the research products, random errors in the CPR are considered based on the observation window, and noise in ATLID signals are added using a noise simulator. We consider the spectral misalignment effect of the visible and near-infrared MSI channels based on response functions depending on the angle from nadir. We discuss plans for updating JAXA EarthCARE synthetic data using a large eddy simulation and implementation of a three-dimensional radiation model.

## 1 Introduction

The Earth Clouds, Aerosol, and Radiation Explorer (EarthCARE) satellite is a joint mission of the Japanese Aerospace Exploration Agencies (JAXA) and the European Space Agency (ESA) (Illingworth et al. 2015, Wehr et al. 2023). The satellite will carry four instruments: a 94 GHz cloud-profiling radar (CPR), a 355 nm atmospheric lidar (ATLID), a seven-channel multispectral imager (MSI), and a broadband radiometer (BBR). These instruments are aboard a single platform and are expected to provide synergistic products. Nominal level 1 (L1) data are observed directly by the instruments. There are plans to produce retrieval products (L2) for clouds, aerosol, and radiative properties using L1 data from single or multiple instruments. For the development and validation of L2 data, pre-launch simulated L1 data are required. The JAXA EarthCARE-like L1



synthetic data (JAXA L1 data) were developed using a global storm-resolving model (GSRM; Satoh et al., 2019; Stevens et al., 2019) and a satellite simulator developed by JAXA and The University of Tokyo, Japan .

The simulation scenes for JAXA L1 data were constructed using numerically simulated GSRM data, which resolve cloud and precipitation systems without global convective parametrization by enhancing horizontal resolution above that of a typical global circulation model (GCM). One of the merits of GSRM is that it does not heavily rely on ambiguous assumptions of cloud fractions at subgrid scales, in contrast to GCM. As pioneering studies of GSRMs, the Nonhydorstatic Icosahedral Atmospheric Model (NICAM; Satoh et al., 2014) has been evaluated and improved using various satellite data (e.g., Masunaga et al., 2008; Roh and Satoh, 2014, 2018; Roh et al., 2017, 2020). Evaluations have included global precipitation and cloud systems in various locations.

A satellite simulator is a collection of radiative transfer models used to simulate satellite-like signals based on outputs of atmospheric models such as GSRMs and GCMs (e.g., Bodas-Salcedo et al., 2011; Hashino et al., 2013, 2016; Matsui et al., 2014; Saunders et al., 2018). Simulators have been developed to evaluate, improve, and compare numerical models using satellite observation data. Here, the Joint Simulator for Satellite Sensors (Hashino et al. 2013, 2016; Satoh et al. 2016) was used as a satellite simulator to produce EarthCARE synthetic data before the launch of the satellite.

JAXA L1 data have been used in several studies to evaluate the performances of CPR and MSI. Hagihara et al. (2021) investigated expected Doppler errors based on the instrument settings of the observation window. In testing different observation windows of CPR, it has been found that the unfolding correction and increased horizontal sampling reduced Doppler errors. The latitude variation of Doppler errors has also been investigated using JAXA L1 data (Hagihara et al., 2022), and Wang et al. (2022) investigated the SMILE (Spectral Misalignment Effect) of MSI data on the cloud retrieval algorithm.

JAXA L1 data can also be used as a testbed to check retrieval algorithm performance, and it is possible to directly compare original cloud data and precipitation simulated by GSRM with data retrieved from retrieval algorithms for each sensor.

JAXA L1 data are of two types, namely the standard product and the research product, the latter of which includes noise and more detailed information about instrument settings. The former comprises two sets of orbit data covering two full global circles, whereas the latter has shorter frames with more detailed instrument settings for retrieval algorithm developers.

Here we introduce JAXA L1 data. Detailed information concerning input data, the orbit/scan simulator, and satellite simulators are described in Section 2. Data for each sensor are also described with instrument settings and output data. Recent and planned developments are discussed in Section 4, including a large eddy simulation and three-dimensional (3D) radiation model.

## 2 Data and model descriptions

### 2.1 Global storm-resolving simulations

JAXA L1 data are based on input data for meteorological conditions, distributions and characteristics of clouds, precipitation, and aerosols related to signals from satellite sensors. We used numerically simulated data from the Nonhydrostatic ICosahedral



Atmospheric Model (NICAM; Tomita and Satoh, 2004; Satoh et al., 2008, 2014) as a GSRM. Horizontal resolution was ~3.5 km and the vertical grid had 40 levels (table 1 in Satoh et al., 2010). The simulation commenced at 00:00Z on 15 June 2008

and was initialized using a 0.5° × 0.5° ECMWF (European Centre for Medium-Range Weather Forecasts) Year of Tropical Convection analysis (Waliser et al., 2012); data for 00:00Z on 19 June, 2008, were used here. A bulk single-moment cloud microphysics scheme with six water categories (NSW6; Tomita 2008) and the MYNN2 (Nakanishi and Niino, 2009) boundary-layer scheme were applied. See previous works (Hashino et al. 2013; Yamada et al. 2016; Nasuno et al. 2016) for the details of the simulation data. The data have been analyzed also in several papers (Hashino et al. 2016; Matsui et al. 2016; Roh et al.

2017; Kubota et al. 2020).

Aerosol data were simulated using the NICAM Spectral Radiation–Transport Model for Aerosol Species (NICAM–SPRINTARS; Takemura et al., 2000), which was implemented using a global 3D aerosol transport–radiation model. The horizontal resolution was ~240 km, and the vertical resolution was the same as that used in the 3.5 km mesh simulation. Aerosol data simulated by NICAM–SPRINTARS include carbonaceous aerosols (black carbon and organic matter), sulfate,

soil dust, sea salt, and the precursor gases of sulfate (sulfur dioxide and dimethylsulfide (DMS)). Aerosol data were used with the ATLID, MSI, and BBR simulations.

The relationship between orbits and cloud distribution in the NICAM simulation is shown in Fig. 1, where simulated 11 μm brightness temperatures (representing cloud top temperatures) indicate high clouds. Lines indicate expected EarthCARE satellite orbits.

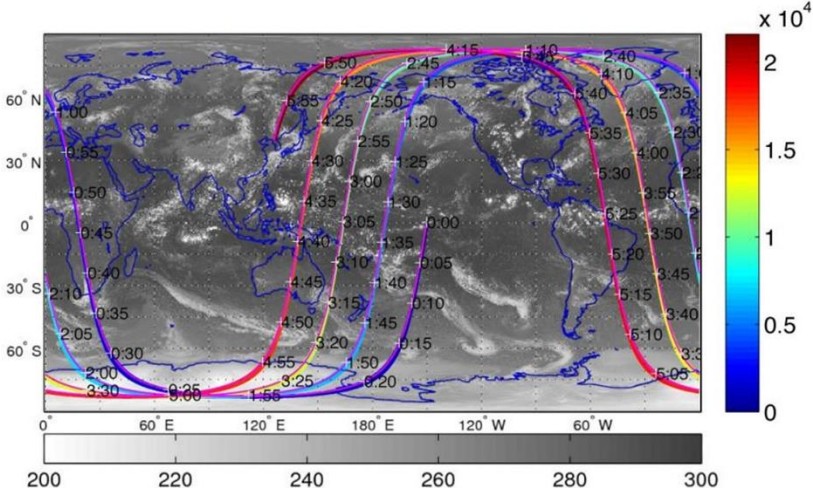


**Figure 1.** Simulated tracks and swath of the EarthCARE satellite. The black/white contour is the 11 μm brightness temperature (K). Colors indicate the time from the starting point (00:00Z) in seconds.





## 2.2 Joint Simulator for satellite sensors

The Joint Simulator for satellite sensors (Hashino et al., 2013, 2016) was used to simulate JAXA L1 data from NICAM data.
The Joint-Simulator was developed as part of the JAXA EarthCARE mission (Satoh et al. 2016) and has simulators for a
visible/infrared imager, radar, lidar, and broadband radiometer corresponding to MSI, CPR, ATLID, and BBR EarthCARE
sensors. It also has a microwave radiometer simulator. The basic structure was inherited from the Satellite Data Simulator Unit
(SDSU; Masunaga et al., 2010) and the NASA Goddard SDSU (Matsui et al., 2014); several simulators with these SDSUs
were shared. The Joint-Simulator has a history of evaluations and improvements of NICAM (Hashino et al., 2013, 2016; Roh
et al., 2020). The settings and descriptions of the simulators are described for each sensor in Section 3.

## 2.3 Orbit/scan simulators

Orbit/scan simulators produce orbit and swath data based on EarthCARE and NICAM data; the orbit simulator determines
the satellite location and the scan simulator determines sampling intervals and the maximum sample number per scan. The
simulators are described in Matsui (2013).

The orbit/scan simulator assumes a Kepler satellite orbit, and six Keplerian elements are needed to calculate satellite position
including inclination, an argument of perigee, and the right ascension of the ascending node. The satellite is in an elliptical
orbit and eccentricity, the semi-major angle, and orbit inclination angle define the shape and size of the orbit.

The orbit was designed as the EarthCARE passed the equator at 14:00 local time in descending node. For this, we set up a
semi-major axis of 6771.28 km, eccentricity of 0.001283, and an orbit inclination angle of 97.05°, together with initial values
of mean anomalies of 270°, an argument of perigee of 270°, and a right ascension of the ascending node of 297.5°.

The along- and cross-track sensor sampling intervals were 500 m, 285 m, 500 m, and 10 km for CPR, ATLID, MSI, and BBR,
respectively. There were 384 samples per the scan for the MSI, with 102 nadir pixels. The ATLID was considered with 3° of
off-nadir angle, as for CALIPSO. The Joint-Simulator applies vertical interpolation on the NICAM data to obtain the samples
on the vertical grid defined for each sensor.

There are eight frames for an EarthCARE single orbit, A–H, divided at latitudes of 22.5°N/S and 62.5°N/S. For example,
Frame A spans from 22.5°S to 22.5°N in ascending mode. Data for two EarthCARE orbits were the standard product.



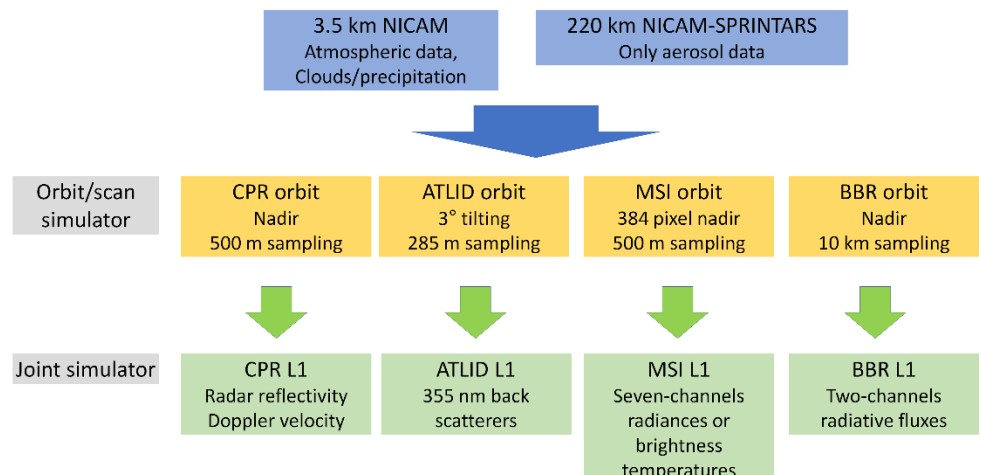

**Figure 2.** Flowchart for production of JAXA L1 data.

## 3 Simulation of EarthCARE signals

A flowchart describing the production of JAXA L1 data is shown in Fig. 2. Input data for the Joint-Simulator were provided by the orbit/scan simulators based on numerical data of NICAM and NICAM–SPRINTARS. Input data were provided for each instrument with the same horizontal resolution and frames. More output JAXA L1 data are used for the validation of L2 products, as described below.

### 3.1 CPR

CPR is a 94 GHz cloud profiling radar that can detect radar reflectivity and Doppler velocities. The minimum radar reflectivity is −36 dBZ, which is a higher sensititvity than that of CloudSat because of the larger antenna and lower orbit than CloudSat.

Radar reflectivity and Doppler velocity were simulated by the EarthCARE Active Sensor Simulator (EASE; Okamoto et al., 2007, 2008; Nishizawa et al., 2008). The EASE simulator takes into account the attenuation of radar related to water vapor and hydrometeors. Doppler velocity is calculated using the terminal velocity of hydrometeors weighted by the radar reflectivity and air motion. We set the vertical resolution of CPR at 99.9308 m. The lowest altitude is 50 m, and the top of the observation window is 19936.23 m. We added a total extinction coefficient of 94 GHz.

The CPR data of the JAXA L1 data are shown in Fig. 3, crossing over the African continent with Frame A. Convective clouds are located near the equator, and a high fraction of cirrus clouds is present in this frame. The 94 GHz radar reflectivity is sensitive to both cloud and precipitation particles. However, the attention by liquid hydrometeors is higher than that of the precipitation radar of the Global Precipitation Measurement (GPM) system. Doppler velocity is not affected by the attenuation.



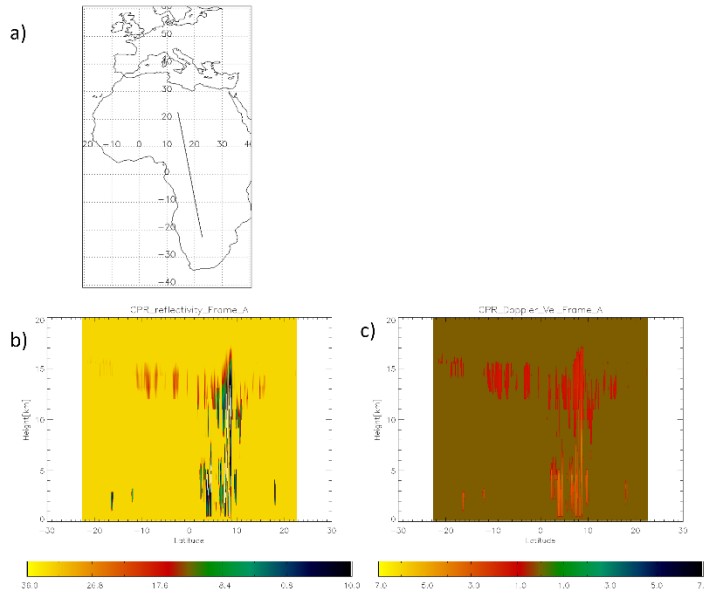

**Figure 3.** CPR data of the JAXA L1 data for Frame A over the African continent **(a)**, with radar reflectivities **(b)** and Doppler velocities **(c)**.

Surface clutter was considered as being based on the response function of CPR. The EarthCARE CPR has less surface clutter than CloudSat; an example of expected surface clutter over the ocean shown in Fig. 4. It is possible to detect low clouds higher

than 600 m. Expected surface clutter was calculated to indicate the limitation of low-cloud height and to provide realistic JAXA L1 data. Here, the normalized radar cross-section of the surface was set to 10 dB over the ocean and 0 dB over the land. An example of surface clutter over the ocean in the JAXA L1 data is shown in Fig. 5. The normalized radar cross-section over oceans depends on surface winds and sea surface temperature; that over land is more complicated that over oceans and will be updated after the EarthCARE satellite launch.

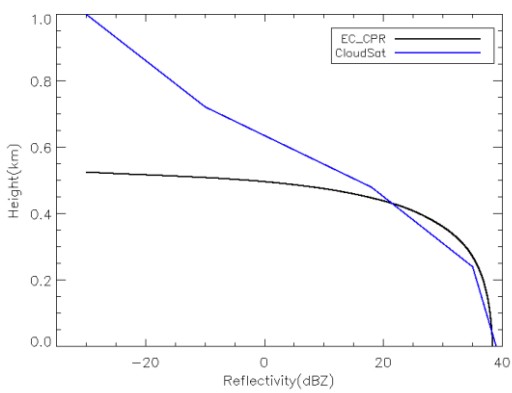

**Figure 4.** Radar reflectivities due to surface clutter of CPR compared for EarthCARE and CloudSat over the ocean.



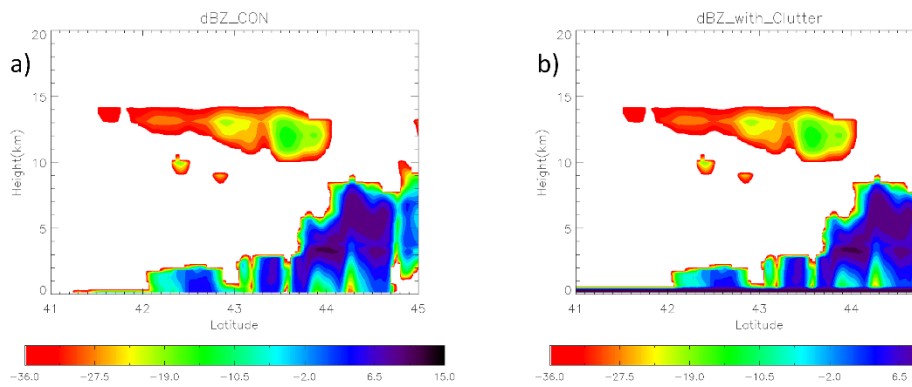

**Figure 5.** Radar reflectivity without **(a)** and with **(b)** surface clutter over the ocean, based on the response function of CPR.

The observed Doppler velocity from the EarthCARE CPR would be expected to have random errors because of the slight vibration of the instrument on the satellite. The maximum Doppler velocity and its random errors were determined by setting the pulse repetition frequency (PRF) of the CPR. CPR has two modes of observation window in operation: low mode (−1 to 16 km) at latitudes of 60°–90° and high mode (−1 to 20 km) at latitudes of 0°–60° (Hagihara et al., 2022). The other alternative observation window is the middle mode between −1 and 18 km. The PRF changes in a range of 6100–7500 Hz with latitude and observation window because the PRF is determined by the satellite altitude (fig. 1 of Hagihara et al. (2021). The observation window setting is based on that of Hagihara et al. (2021), who investigated random errors of Doppler velocity of CPR in different modes.

The Doppler velocity of an example of the JAXA L1 data is shown in Fig. 6 for observation windows of 16, 18, and 20 km. The 20 km window mode reproduced the noisy Doppler velocity for ice and rain (Fig. 6b) relative to the original L1 data. This mode has a small range of minimum–maximum Doppler velocity, and it is particularly difficult to retrieve the terminal velocity of ice particles in cirrus clouds. The 16 km observation window yields better performance of Doppler velocity for ice and rain than the other two windows (Fig. 6d), and the top of cirrus clouds is located near an altitude of 15 km. However, it is still possible to neglect high clouds above 16 km over tropical regions. This indicates the need for care in evaluating the Doppler velocity of ice clouds using a 20 km observation window. The uncertainty in Doppler velocity can be derived from the observation window using the Joint-Simulator.





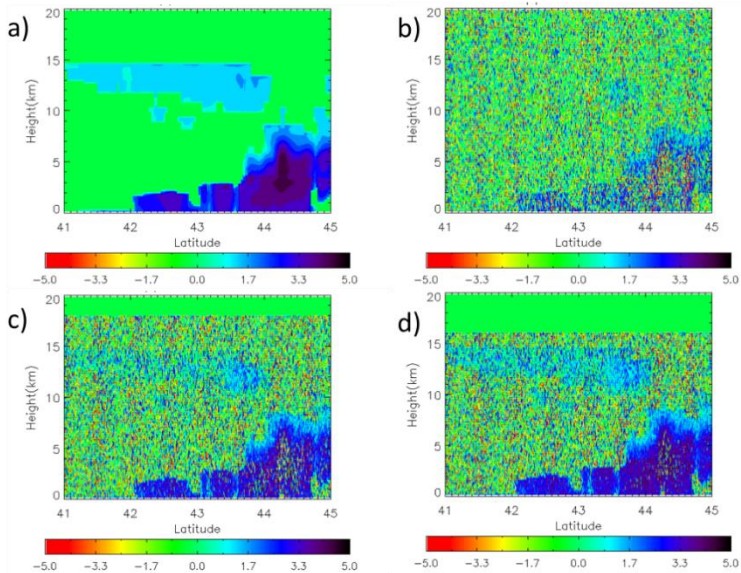


**Figure 6.** Examples of Doppler velocity for the original L1 data using a 20 km observation window (**a**), high mode (**b**), 18 km observation window and middle mode (**c**), and 16 km observation window and low mode (**d**).

### 3.2 ATLID

ATLID is the 355 nm high-spectral-resolution lidar, which can observe 355 nm backscatter from Mie and Rayleigh scattering.

The Mie scattering channel with co-polarization is related to cloud and aerosol particles, and the Rayleigh scattering channel with co-polarization is related to atmospheric molecules. The total attenuated backscatter channel with cross-polarization is related to the shapes of hydrometeors and aerosols.

EASE simulates the lidar signals of ATLID by considering the scattering and attenuation of molecules, hydrometeors, and aerosols. The outputs of ATLID are 355 nm total attenuated backscatters from Mie or Rayleigh scattering. The effect of

multiple scattering by liquid hydrometeors on lidar signals was considered using a correction factor parameterized using Monte Carlo simulation (Ishimoto and Masuda, 2002). We provide CALIPSO lidar signals of 532 nm with a depolarization ratio of 532 nm. The parameterization of the depolarization ratio is described by Roh et al. (2020).

Signals of ATLID data over the African continent for Frame A (Fig. 3a) are shown in Fig. 7. The attenuation of water clouds is more pronounced than that for CPR below 5 km height. The Rayleigh channels show the backscatter from atmospheric

molecules (Fig. 7b), and the Mie channels show cloud and aerosol distributions (Fig. 7c, d). Saharan dust is located within 10°N and 20°N (Fig. 7c, d). For the validation of retrieval algorithms, 355 nm extinction coefficients are provided for liquid and ice clouds, dust, sulfate, sea salt, and black carbon/organic carbon, as well as the molecular extinction coefficient.



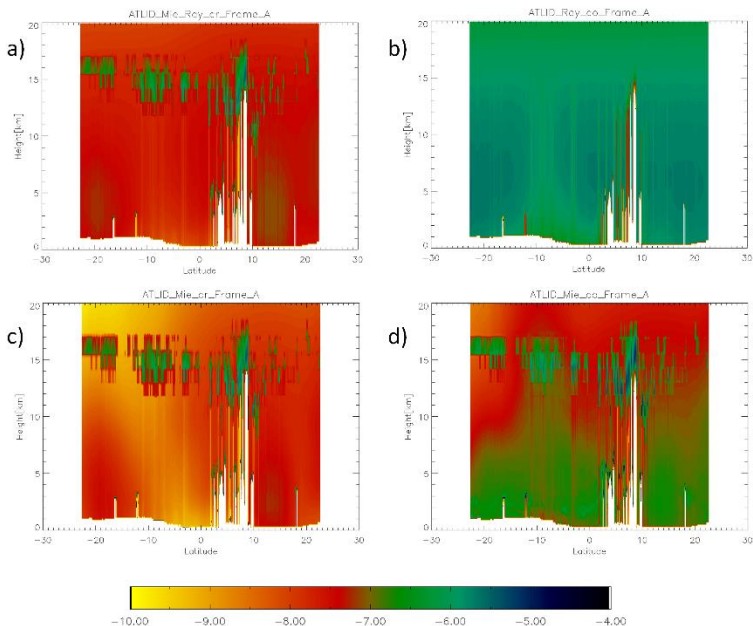

**Figure 7.** Examples of ATLID L1 data for Frame A (Fig. 3a), showing combined Rayleigh (Ray) and Mie channels with cross-polarization
(CR) **(a)**; Rayleigh channels with co-polarization (CO) **(b)**; and Mie channels with CR **(c)** and CO **(d)**. Contours are shown on a log scale of
the backscattering coefficients ($m^{-1}$ $str^{-1}$).

For realistic ATLID L1 data, possible random noise was also considered in the simulator, with noise data provided by ESA.
The noise model was based on a Gaussian random noise from shot noise, dark count rate, and solar background counts of
ATLID.

Examples of ATLID signals with and without noise are shown in Fig. 8, with two cloud layers related to cirrus clouds above
10 km and water clouds below 10 km. There is strong attenuation in the water clouds, under which values are undefined (Fig.
8a). The undefined values are filled with random noise in the simulation, and it is possible to mis-classify the area under the
cloud. Using this random noise, the retrieval algorithm developer could consider the expected random noise when retrieving
physical variables related to aerosols and clouds.





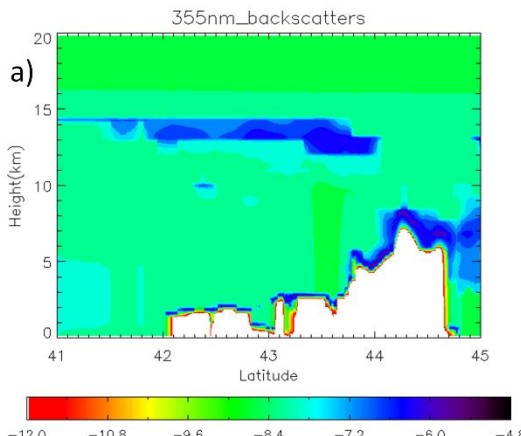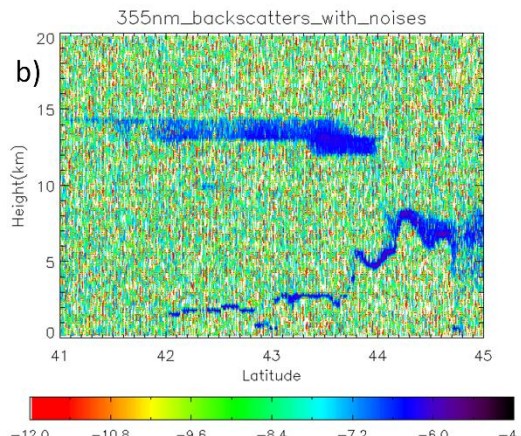

**Figure 8.** Examples of 355 nm total attenuated backscattering coefficients of ATLID without **(a)** and with **(b)** random noise. The contour is a log scale of backscattering coefficients (m$^{-1}$·str$^{-1}$).

### 3.3 MSI

MSI is a passive sensor used to observe infrared and reflected solar radiances, with seven channels at 0.67, 0.865, 1.65, 2.21, 8.80, 10.8, and 12.0 μm. The total pixel number is 384 in the direction orthogonal to the satellite orbit. The approximated nadir location is in the 102$^{nd}$ pixel. MSI signals were calculated by RSTAR (System for Transfer of Atmospheric Radiation; Nakajima and Tanaka, 1986, 1988) as the sensor simulator. RSTAR (Nakajima and Tanaka, 1986) derives the solution of the discrete-ordinate method using Eigenspace transformations of symmetric matrices. RSTAR is a general package for simulating radiation fields in the atmosphere–land–ocean system at wavelengths of 0.17–1000 μm. Monochromatic intensity and intensity with a finite range of wavelengths can be calculated, as required for channels with significant gas absorption. Three streams were set in each hemisphere in the Joint-Simulator (i.e., the six-stream method).

We used a fixed wavelength for each channel as the default setting in the JAXA L1 data. The units of channels are radiances [W·m$^{-2}$·str$^{-1}$·μm$^{-1}$] for 0.67, 0.865, 1.65, and 2.21 μm. The unit for the other channels is brightness temperature (K). Optical depths of clouds and aerosols were calculated to validate the algorithms. Examples of MSI L1 data for seven channels over the ocean are shown in Fig. 9 for Frame F. High clouds are located near latitude 33°S, with strong reflection in the 0.67 μm channel and low brightness temperatures in the 8.80, 10.8, and 12.0 μm channels.

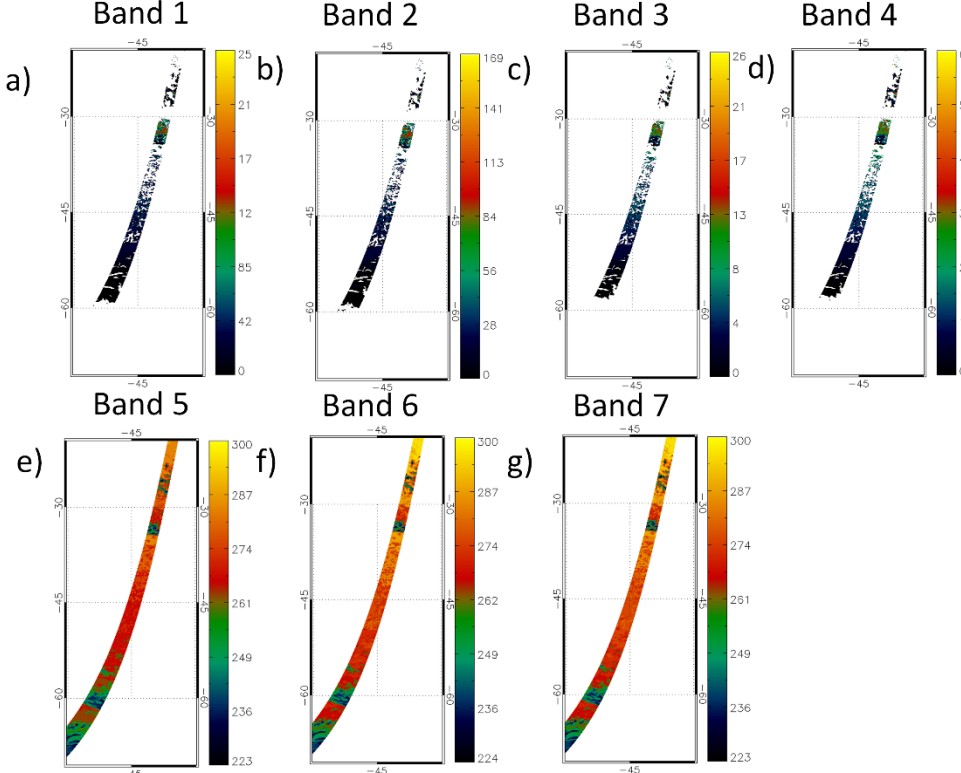

**Figure 9.** Examples of MSI data over the ocean for Frame F for seven channels at 0.67 **(a)**, 0.865 **(b)**, 1.65 **(c)**, 2.21 **(d)**, 8.80 **(e)**, 10.8 **(f)**, and 12.0 μm **(g)**. The contours of the upper panels are radiance ($W \cdot m^{-2} \cdot str^{-1} \cdot \mu m^{-1}$) and those in the bottom panels are temperatures (K).

MSI has a spectral distortion termed the "smile effect", which can be considered using the shifted response function in the spectral domain, depending on the across-track pixel in the swath. MSI is known to be affected by the smile effect in the 0.67, 0.865, 1.65, and 2.21 μm channels. The shift of wavelength in the 0.67 and 1.65 μm channels is more obvious than that in the 0.865 and 2.21 μm channels.

We introduced the response function of MSI to reproduce the smile effect in the Joint-Simulator and simulated two channels of MSI in the research product. We investigated its effect on the radiance of the 0.67 and 1.65 μm channels. Two sets of the shifted response functions for the smile effect for these channels are shown in Fig. 10. These channels are used to retrieve cloud properties; the 0.67 μm channel is primarily sensitive to cloud optical thickness and the 1.67 μm channel to cloud effective radius (e.g., Platinick et al., 2017).

Differences between the fixed response function on the nadir and that with the smile effect for the descending scene of the satellite are shown in Fig. 11. For the 0.67 μm channel, there are large differences on the left side of the satellite direction, with signals with the smile effect being underestimated relative to the simulation using the fixed response function. The





difference is greatest at the edge of the swath with spectral distortion, where the maximum difference is 5.76 radiances (W·m$^{-2}$·str$^{-1}$· µm$^{-1}$). The difference in the right half of the swath is not greater than that on the other side.

For the 1.65 µm channel, the differences in radiance are smaller than those of the 0.67 µm channel. The smile effect causes
a positive bias on the right and a negative bias on the left of satellite direction in the 1.65 µm channel, with a different pattern to the 0.67 µm channel. Wang et al. (2022) investigated the smile effect for the cloud retrieval algorithm for water and ice clouds over the ocean using these MSI data.


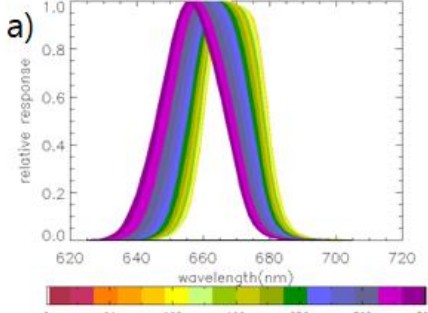
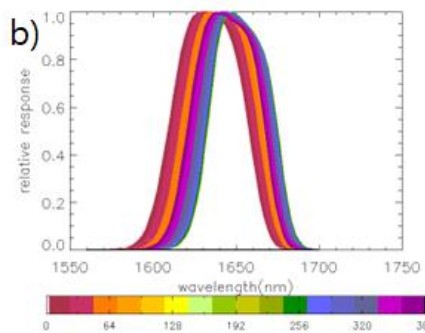

**Figure 10**. Response functions of MSI (considered the smile effect) for the 0.67 µm channel (Band 1) **(a)** and the 1.65 µm channel (Band 3) **(b)**. Colors indicate MSI pixel numbers.


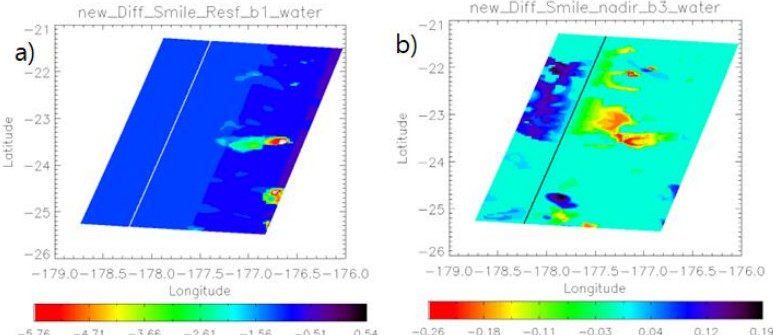

**Figure 11.** Examples of differences between MSI signal with a fixed response function and with the smile effect for the 0.67 µm (Band 1) **(a)** and 1.65 µm (Band 3) **(b)** channels over the ocean, for the descending mode where the satellite is moving southwestward.






### 3.4 BBR

BBR has two channels of 0.25 and 50 μm for observing the total radiative flux at the top of the atmosphere (TOA) and 0.25 and 4 μm for the short-wave radiative flux at TOA with 10 km horizontal sampling. The long-wave flux at TOA is obtained

by subtracting the short-wave flux from the total flux. There are three view modes of nadir, forward, and backward in the BBR. Only the nadir mode was calculated for the JAXA L1 data.

Radiative fluxes were simulated by MSTRN-X (Sekiguchi and Nakajima, 2008). MSTRN-X uses two-stream approximation and the correlated k-distribution (CKD) methods to model gas absorption using quadrature points and weights. MSTRN-X considers 28 species compiled in HITRAN2004, and the radiative transfer solver uses a two-stream approximation. MSTRN-

X is also used as the radiation scheme for NICAM simulations (Satoh et al., 2014). We did not consider the radiative effect of aerosols in the NICAM simulation, so we calculated aerosol transport using NICAM–SPRINTARS with coarse resolution. BBR data are produced using simulated data with aerosols.

BBR data comprise short-wave and long-wave fluxes with downward and upward directions at TOA and the surface; these data are to be used for validation. We also added vertical profiles of short-wave/long-wave heating rates with 500 m vertical

resolution, and the optical depth at 532 nm.

BBR data are intended to be used to evaluate the radiative transfer calculation from retrieved products of vertical profiles of clouds and aerosols. Examples of BBR data and their relation to the CPR signals in JAXA L1 data are shown in Fig. 12. Multi-layer clouds were located in the southern part of the orbit between 32°S and 30°S (Fig. 12a, b), where the outward short-wave fluxes at TOA are large due to the multi-layer clouds having stronger reflectance than cirrus clouds. Short-wave heating in

upper cloud layers and long-wave heating/cooling near the cloud layers are reasonably simulated (Fig. 12c, d). Note that the horizontal resolution of CPR is 500 m and differs from the resolution of the BBR signals, where the short-wave/long-wave heating rates are calculated with a 10 km resolution.

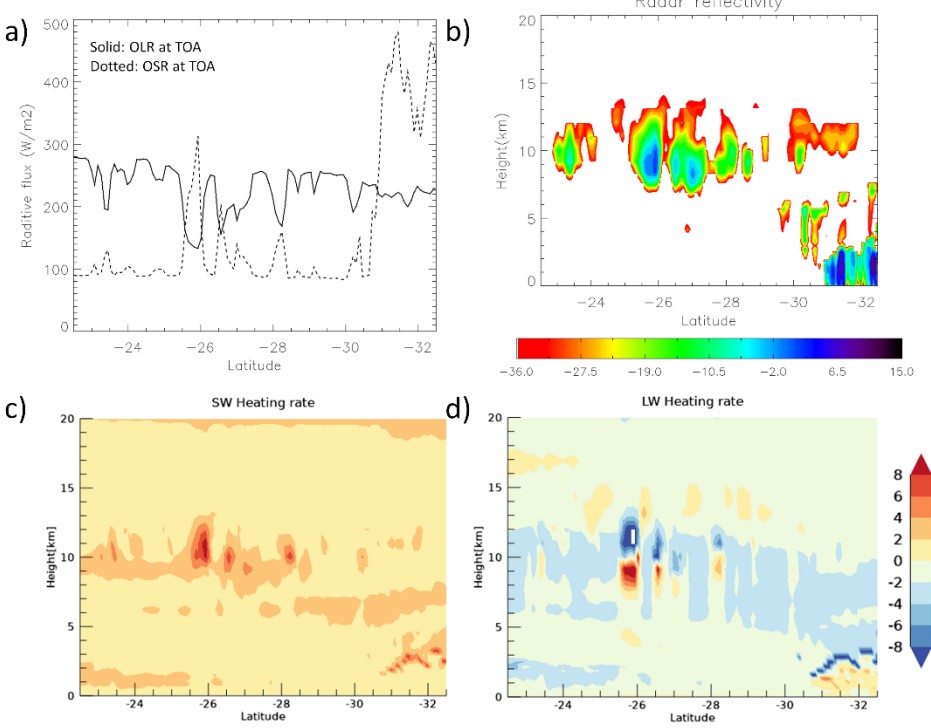

**Figure 12.** Examples of BBR simulations **(a)** and corresponding CPR signals **(b)**. Short-wave **(c)** and long-wave **(d)** radiative heating rates are also provided as JAXA L1 data for use in validation.

## 4 Discussion

We have introduced JAXA L1 data that are currently distributed to L2 algorithm developers. Although these data are useful, we have plans for improvement, as discussed below.

### 4.1 High-resolution experiments

The movement of the EarthCARE satellite affects the Doppler velocity measurement by CPR. This means there are more variations in Doppler velocity in horizontal sampling as, for example, with positive Doppler velocity in the forward part of horizontal sampling and negative velocity in the backward part. The weighting of Doppler velocity in horizontal sampling differs with the shape of the antenna, and non-uniform beam filling effects (NUBFs) on the accuracy of Doppler velocity should be investigated and means found for its improvment.

The NICAM simulation was undertaken using a 3.5 km horizontal resolution over the global domain for JAXA L1 data. However, simulation data with higher resolution than the horizontal CPR sample are desirable for L2 algorithm developers,





and not necessarily for the global domain. Therefore, we undertook a regional high-resolution simulation using ASUCA (A System based on a Unified Concept for the Atmosphere; Ishida et al., 2022). ASUCA is a regional operational model of the Japan Meteorological Agency (JMA). We conducted a simulation using 100 m horizontal resolution with ASUCA over the Kanto area within the ULTIMATE (ULTra-sIte for Measuring Atmosphere of Tokyo metropolitan Environment) research project (Satoh et al., 2022). We prepared three cases in September 2019, covering an intensive observation period by cloud

radar from ground level.

Horizontal distributions of precipitation between ground radar observations and the ASUCA simulation are shown in Fig. 13 for Typhoon Faxai. ASUCA reproduced detailed structures of rain bands similar to observations. The Joint-Simulator simulated the cross-section of radar reflectivity and Doppler velocity of EarthCARE (Fig. 13c, d). The higher-resolution experiment reproduced a more detailed structure of the eyewall system of the cyclone, but the domain size was limited by

computational resources. These data were used in the production of new JAXA L1 data to investigate NUBFs of CPR as the research product. We evaluated the ASUCA simulations using intensive ground observations, data from which are also helpful in validation of the EarthCARE satellite after launch.

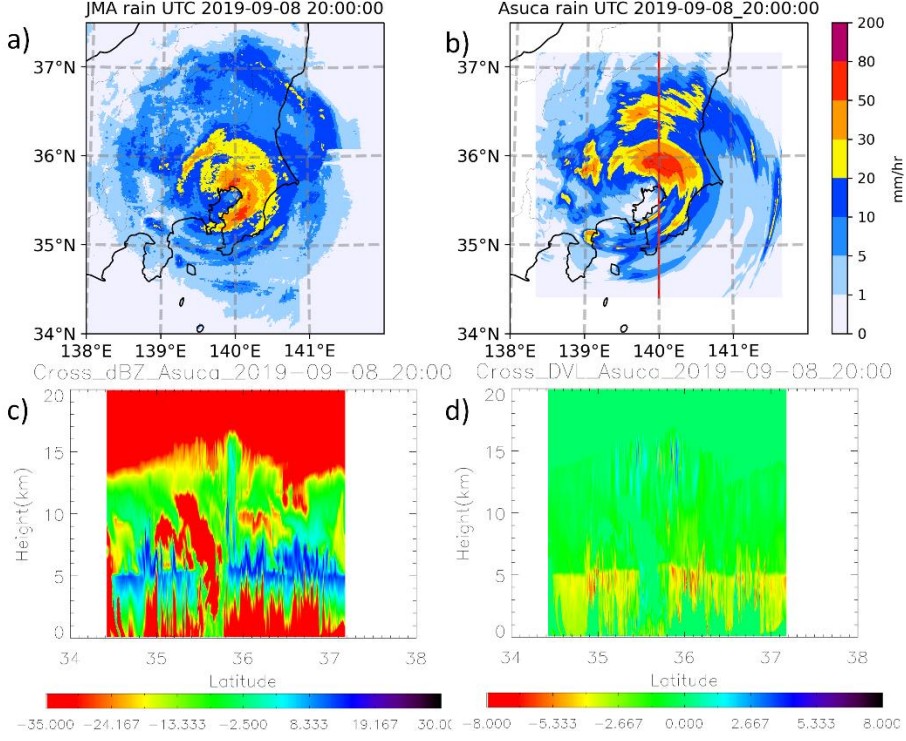


**Figure 13.** Example of the ASUCA simulation for Typhoon Faxai, with horizontal distributions of observed precipitation **(a)**, the ASUCA simulation **(b)**, cross-sections of radar reflectivity **(c),** and Doppler velocity **(d)**. (c) and (d) pertain to the red line in (b).




## 4.2 3D radiation

In the EarthCARE research product, 3D cloud fields will be constructed using the three sensors CPR, ATLID, and MSI. Currently, JAXA L1 data are based on a 1D radiation calculation, which has limitations in representing scattering properties in 3D cloud fields. For study of the effect of 3D fields, 3D radiation calculations are needed. Okata et al. (2016) developed the 315 3D radiation model (MCstar) using a Monte Carlo method and investigated the 3D radiation effect of 3D cloud fields constructed by CPR of CloudSat and Moderate Resolution Imaging Spectroradiometer/Aqua data on the A-train. We plan to implement MCstar in the Joint-Simulator and produce new MSI and BBR L1 data based on the 3D radiation calculation with high-resolution simulation data. Input data of MCstar will be the ASUCA simulation with 100 m horizontal resolution.

## 5 Summary


JAXA EarthCARE synthetic data (JAXA L1 data) were compiled using the global storm-resolving model (GSRM) NICAM simulation with 3.5 km horizontal resolution, and the Joint-Simulator. JAXA L1 data are intended to support the development of JAXA retrieval algorithms for the EarthCARE sensor before launch of the satellite. The expected orbit of EarthCARE and 325 horizontal sampling of each sensor were used to simulate the signals. EarthCARE has four instruments: a 94 GHz Cloud profiling radar (CPR), a 355 nm Atmospheric lidar (ATLID), a seven-channel Multispectral imager (MSI), and a Broadband radiometer (BBR).

CPR is the first atmospheric radar in space with Doppler capability. It has better radar sensitivity with a larger antenna than the previous CPR aboard CloudSat. JAXA L1 data are considered using the same vertical and horizontal sampling as CPR. 330 Surface clutter over both the ocean and land was added based on the response function of CPR in the research product. Expected random errors in Doppler velocity were considered based on three observation windows (Hagihara et al., 2021).

ATLID is the 355nm high-spectral-resolution lidar with three channels: the Mie channel with co-polarization, the Rayleigh channel with co-polarization, and the total channel (Mie + Rayleigh channels) with cross-polarization. JAXA L1 data include these three data channels related to clouds, aerosols, and atmospheric molecules. The Mie channel with co-polarization is 335 related to clouds and aerosols, and the Rayleigh channel to scattering from atmospheric molecules. For validations, extinction coefficients were separated between clouds and aerosols. The 534 nm backscatter was also simulated for comparison with CALIPSO. ATLID data were considered with random noise based on instrument settings for the research product.

MSI is the multi-spectral imager for the observation of emitted infrared and reflected solar radiances and is used to construct 3D cloud scenes using two active sensors. MSI data are calculated using fixed-wavelength data as default data. We investigated





the smile effect using the research product for 0.67 and 1.65 μm wavelengths, with data based on the response function depending on pixel number.

BBR is a multi-angle broadband radiometer with three telescopes with nadir, forward, and backward modes. MSI test data have only the nadir mode and are simulated by the same radiation code as that of NICAM, with 10 km horizontal sampling. Optical depth and vertical profiles of heating rate for shortwave and longwave radiation were added.

We have plans for improvement of the JAXA L1 data for the investigation of NUBFs on the accuracy of Doppler velocity. Higher-resolution data are required for the distribution of Doppler velocities with <500 m horizontal sampling size. We undertook regional simulations with 100 m horizontal resolution over the Kanto region of Japan to produce new JAXA L1 data. To investigate the 3D radiation effect, we will add a 3D radiation model to the Joint-Simulator and introduce MSI and BBR data for 3D radiation to the research product.

After the launch of EarthCARE, its data will provide new insights for the evaluation and improvement of GSRMs. According to the first GSRM intercomparison study, vertical profiles of cloud ice and water vary among models, although the horizontal distribution of OLR is consistent (Roh et al., 2021). EarthCARE can provide more detailed information on the vertical distribution of hydrometeors, with two active sensors of CPR and ATLID for the validation of GSRMS.

The production of JAXA L1 data is related to the development of the Joint-Simulator, which has been updated with detailed
settings of instrument information for EarthCARE. These updates will improve our understanding of uncertainties in observations and retrieved values. Roh et al. (2022) compared two microphysics schemes using CPR on the ground and the expected CPR of EarthCARE using the Joint-Simulator and found the expected Doppler velocity of EarthCARE from the low-window mode would be better for evaluating the characteristics of cloud microphysics schemes consistent with ground observation data.


*Data availability*

We can share EarthCARE synthetic data. Please email ws-roh@aori.u-tokyo.ac.jp if interested. The Joint-Simulator is available from https://www.eorc.jaxa.jp/theme/Joint-Simulator/userform/js_userform.html.


*Author contributions*

WR drafted the manuscript and produced the JAXA L1 data. MS checked the manuscript and helped with the production of JAXA L1 data. HT developed the Joint-Simulator and supported JAXA L1 data production. SM did the ASUCA simulations. TN did a global storm-resolving simulation data using NICAM. TK led the Joint-Simulator development and provided
feedback on the manuscript draft.





*Competing interests*

The authors declare that they have no conflicts of interest.

*Acknowledgements*

The authors thank members of the JAXA EarthCARE Science Team and the Joint-Simulator project. The authors also thank
to ESA for providing the measured value of response functions of EarthCARE/MSI. The authors thank Dr. Toshi Matsui for
providing the orbit/scan simulator. Computational resources were partly provided by the National Institute for Environmental
Studies.

*Financial support*

This work was supported by the EarthCARE satellite study commissioned by the Japan Aerospace Exploration Agency. MS
and WR were supported by a Grant-in-Aid for Scientific Research B (20H01967) and the Program for Promoting
Technological Development of Transportation of the Ministry of Land, Infrastructure, Transport, and Tourism (MLIT).

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
