# Peer review of "Introduction to EarthCARE synthetic data using a global storm-resolving simulation"

_Atmospheric Measurement Techniques, 2023_

## Author Response (AR1)

To Reviewer 1:

We appreciate the reviewers' constructive comments. We have revised our manuscript accordingly, and we hope the reviewer will find the revisions satisfactory.

**Review of "Introduction to EarthCARE synthetic data using a global storm-resolving simulation" by Roh et al., submitted to Atmospheric Measurement Techniques (AMT)**

[Article#: amt-2023-18]

This report contains general, major, and specific comments from this reviewer on the manuscript.

**A summary of the manuscript and general assessment:**

Recommendation: Major revision

This manuscript introduces synthetic level-1 (L1) data products of the EarthCARE space-borne instruments, which will be launched in a few years. The purpose of producing the data is to support the development of retrieval algorithms to create higher-level data products before the satellite observations start and the operational product release. The synthetic data was produced by simulating the satellite orbits and observations in the geophysical fields of global storm-resolving model simulation with a 3.5 km horizontal mesh. In addition, the distribution of aerosol concentrations, which was not included in the storm-resolving model simulation, was considered by implementing additional global aerosol transport simulation with a coarser horizontal mesh. The data products include standard one with two full orbits around the earth and research-mode one with a part of the orbits adding corrections in the satellite simulations, which imitate the observation products by the actual instruments more realistically.

The scope of the manuscript is within the main subject areas of AMT, specifically theoretical calculations of measurement simulations with detailed error analysis, including instrument simulations.

I suggest a major revision. I have no deep expertise in the measurements using each remote-sensing instrument. However, from such a perspective, the current manuscript needs to be improved for better readability and clearer points. In addition, the present descriptions of the data availability could be better because this manuscript aims to introduce and advance the

use of the EarthCARE synthetic data by other engineers and researchers for retrieval algorithm development. I list major problems in the following section.

**Major comments:**

1. Abstract

The abstract needs to be rewritten to meet the objectives of the manuscript. At least, related to the data availability, the information about how to get the data sets is necessary for the abstract.

→ Our data is open to the public. We added information about the data availability in the abstract. We uploaded the standard product data in the zenodo (https://zenodo.org/record/7835229) and introduced the location in the data availability section.

2. Figure quality

The quality of the figures in the current discussion preprint could be better, although it might be degraded in the preprint production. Some specific comments for each figure can be found in the specific comments section.

→ We improved several figures (Fig. 3, 4, 5, 6, 7, 8, 9, 11, and 13) based on your comments.

3. Difference between the standard and research products

The differences between the standard and research(-mode) products need to be presented in a more explicit format. Adding a table summarizing the differences may be helpful to show what is in or not in each product. The authors present some examples of the synthetic data products in the manuscript. However, which type of product is used to produce the examples and what are the differences in the case which type is used to make the examples need to be clarified.

→ We added the table about the difference between the standard product and the research product.

Table 1. The differences between the standard product and the research product.

| | Additional data in the research product |
|---|---|
| CPR | Random errors of Doppler velocity based on the observation window, surface clutters |
| ATLID | Random noises |
| MSI | Consideration of the response function depending on the pixel number |

**4. Mixture of the NICAM and NICAM-SPRINTARS simulations against L2 validation**

Similar to algorithm development in other previous works, the original geophysical fields in the atmospheric simulations are expected to work as the true value in the validation of the result in applying the retrieval algorithm to the L1 data to get the L2 data. However, in this study, as long as I understand the process, the NICAM simulation with a 3.5 km horizontal mesh itself does not have aerosol direct effects on the atmospheric radiation in the simulated geophysical fields, and the NICAM-SPRINTARS simulation alternatively provides the spatial distribution of aerosol loading only in the process of creating the L1 data. This inconsistency could be an error in the L2 validation in the retrieval algorithms for the L1 data, specifically BBR. The NICAM simulation has no aerosol direct effects, so there is no true value.

➔ We understand that the NICAM simulations with a 3.5 km mesh have no aerosol effects on the radiation and dynamics. We thought the inconsistency was not critical for aerosol retrievals using ATLID (In personal communication with our aerosol retrieval developers). We have a plan to update the JAXA L1 data with the NICAM-SPLINTARS with a 14 km grid. We will check the impact of the inconsistency between NICAM/NICAM-SPLINTARS on the ATLID and BBR using higher-resolution data.

**5. Data availability**

The data sharing needs to be improved, given the scope and objectives of the manuscript. Although the traditional "upon request" may be acceptable in most research articles, this is not the case. AMT suggests the following data policy for handling data sets:

*the deposit of research data (i.e. the material necessary to validate the research findings) that correspond to manuscripts, preprints, or journal articles in reliable FAIR-aligned data repositories that assign persistent identifiers (preferably digital object identifiers (DOIs)). Suitable repositories can be found at https://www.re3data.org/;*

*the proper citation of data sets in the text and the reference list including the persistent identifier. For data sets hosted on GitHub, authors are kindly asked to issue a DOI through Zenodo and include this DOI in the reference list;*

*the inclusion of a statement on how their underlying research data can be accessed. This must be placed in the section "Data availability" at the end of the manuscript before the acknowledgements. If the data are not publicly accessible, a detailed explanation of why this is the case is required (e.g. applicable laws, university and research institution policies, funder terms, privacy, intellectual property and licensing agreements, and the ethical context of the research);*

*the provision of unrestricted access to all data and materials underlying reported findings for which ethical or legal constraints do not apply.*

The authors should follow the guidelines carefully. In addition, detailed information about the data sets, such as data size, data format, etc., should be included in the section or other parts of the manuscript.

➔ We uploaded our data in the zenodo (https://zenodo.org/record/7835229) and added the information in the "Data availability" section.

**Specific comments:**

Line 19: "discuss" should be "introduce".

➔ We modified it based on your comment.

Line 71: I need clarification on the relationship between the NICAM simulation (Line 64) and the NICAM-SPRINTERS simulation. Was the NICAM-SPRINTERS simulation run separately from the NICAM simulation? The aerosol distribution simulated in the NICAM-SPRINTERS simulation was used for the JAXA L1 data only, and it was not used in the NICAM simulation with ~3.5 km grid spacing for calculating the atmospheric radiation (aerosol direct effects) and the aerosol-cloud interaction. Correct?

➔ Yes, NICAM simulation with 3.5 km grid spacing does not include atmospheric radiation and aerosol-cloud interaction.

Line 75: Related to the comment above, if calculating aerosol particle radiative properties needs information on the ambient atmosphere, such as relative humidity, which simulation provides the information?

➔ The only aerosol data in the NICAM SPLRINTARS data are used to simulate the ATLIMD, MSI, and BBR data. We believe that temperature, pressure, and relative humidity affect the aerosol retrievals. We believe the effect is not significant (a few percent) for the ATLID simulation. We have a plan to do the higher horizontal resolution like 14 km of NICAM-SPLTANRS. And we will check the impact on aerosol retrievals and BBR simulations.

Figure 1: Is the color transition of the line consistent with the color bar on the right side of the figure? Purple is not included in the color bar.

➔ The pule line indicates the swath of the MSI. We added the explanation in the figure caption.

Figure 2: Does the 3.5 km NICAM simulation also provide terrain heights and land/ocean/ice surface states, or is this information from other data sources?

➔ NICAM simulations provided terrain heights and land/ocean/ice surface states.

Figure 3: This figure quality is very poor. It should have a higher dpi to be read. I cannot see which side of the color bars is negative. And which is the upward or downward direction positive vertically?

➔ We improved the figure quality.

Figure 7: The figure quality is very poor again.

➔ We improved the figure quality.

Line 229 and Figure 10: Which is the base? I mean, which is subtracted from the other?

➔ The base is the response function on the nadir. The smile effect results are subtracted from the simulation with the response function on the nadir.

Line 279: What this section does is not discussion. It should be renamed "future plans", "future improvement", or something.

➔ We modified "discussion" to "future improvements".

**Grammatical problems:**

Line 34: Remove "global".

➔ We removed the "global".

Line 178: "The attenuation of water clouds" => "The attenuation by water clouds"

➔ We modified it based on your comment.

Line 227: "1.67" => "1.65"

➔ We modified it based on your comment.

To Reviewer 2:

We appreciate the reviewer's constructive comments. We have revised our manuscript accordingly, and we hope the reviewer will find the revisions satisfactory.

General Comments

============================

I recommend that the paper be accepted subject to minor revisions, mainly editorial in nature. As far as the rational for presenting this work goes, I would suggest that the authors also mention that the work presented here also can aid in engaging and educating the wider atmospheric community (e.g. the modeling community) with respect to the expected utility of the observations and their efficient usage.

The quality of all of the figures need to be improved ! They do not appear to be the required 300dpi resolution. In addition, the size of the font used for the axis labels etc.. is too small.

➔ We added your suggestion. "Satellite remote sensing data have been fruitful in understanding clouds and aerosols. However, it is difficult to interpret the radiances or signals from the sensors. Most of the modeling community uses the retrieved product from the satellite data such as precipitation. This study implies how to understand the directly observed signals and their uncertainty from simulations of the specific instruments of the EarthCARE satellite. This research would be helpful for the modeling community to improve and constrain the physical parameter in the model using satellite observations.." in summary.

➔ We improved several figures (Fig. 3, 4, 5, 6, 7, 8, 9, 11, and 13) based on your comments.

==============================

Specific Comments

=================================
========

Abstract

========
Line 10: "Pre-launch simulated satellite data are useful to develop retrieval algorithms and to facilitate the rapid release of retrieval products after launch"

➔ We modified the sentence based on your comment.

Line 11: "Here we introduce the Japanese..."

➔ We modified the sentence based on your comment.

Line 13: "...data were produced corresponding to the four EarthCARE instruments sensors, namely a 94GHz....."

➔ We modified the sentence based on your comment.

Line 19: "We discuss plans for updating the JAXA using large eddy simulation model data and the implementation of a ....."

➔ We modified the sentence based on your comment.

Introduction

Line 28: "expected to provide synergistic retrieval products..."

➔ We modified the sentence based on your comment.

Line 31: "using a Global Storm Resolving Model (GSRM..."

➔ We modified the sentence based on your comment.

Line 35: "One of the merits of GSRMs is that they do not....in contrast of GCMs.

➔ We modified the sentence based on your comment.

Line 36: "The Nonhydrostatic Icosahedral Atmospheric mode (NICAM...) is one of the pioneering GCSRMs and has been evaluated and ....."

➔ We modified the sentence based on your comment.

Line 45: Please explain more clearly what the observation window means. I believe you are referring to the range interval covered by the CPR (which is determined by the PRF). This may not be evident to many potential readers of this paper.

➔ As you know, the PRF determines the radar range, but the radar range is not the same as the satellite altitude. The observation frame is determined by the PRF. The observation frame is called the observation window in the satellite community. I have changed "the top height of the observation (observation window)".

Line 51: Replace "GSRM" by "NICAM." (you are referring to a specific GCSRM here).

➔ We modified the sentence based on your comment.

Line 55: "Here we introduce the JAXA simulated L1 EarthCARE data set."

➔ We modified the sentence based on your comment.

Line 57: "..in Section 4, including the use of large-eddy simulation model data and the implementation of a three-dimension (3D) radiation model."

➔ We modified the sentence based on your comment.

==========

Section 2
==========
Line 61: "The JAXA L1 simulation data are based on..."

➔ We modified the sentence based on your comment.

Line 62: NICAM has already been defined you do not need to do it again. Also, the sentence is awkward, I suggest something like: "We used NICAM data to drive the instrument simulations. NICAM was configured with a horizontal resolution of about 3.5 km...."

➔ We modified the sentence based on your comment.

Line 78: "The lines indicate the expected EarthCARE orbits corresponding to the simulations presented in this paper."

➔ We modified the sentence based on your comment.

Line 107: Replace last sentence by "Standard product L1 simulated data were generated for two EarthCARE orbits."

➔ We modified the sentence based on your comment.

Section 3

===========

Line 115: The last sentence is unclear. Can the authors clarify what they mean here ?

➔ We deleted the last sentence.

Line 118: "The CPR..."

➔ We modified the sentence based on your comment.

Line 124: The last statement is unclear. A total extinction coefficient was added to what ? The simulated data product ? Or, perhaps, the effects of the extinction were added to the simulated

reflectivity ?

➔ We added a total extinction coefficient was added to the simulated data product. We modified the sentence based on your comment.

Line 134: Can you (very) briefly explain why the EarthCARE CPR is expected to have less surface clutter than the CloudSAT CPR ?

➔ Simply speaking, it originated from the improved hardware. When our group checked the pulse response function on the ground, we could expect better surface clutter in space.

Line 156: Please make it clear what the "observation window" is. See also my earlier comment on this issue.

➔ I explained the "observation window" in the introduction based on your earlier comment.

Line 158: "minimum-maximum Doppler velocity" ? Are you referring to the "Doppler folding velocity" (also know as the "unambiguous velocity")) ? If so, please use the correct term and add a basic radar reference.

➔ We changed to "Doppler folding velocity" based on your comment.

Line 161: The last sentence is unclear. Either explain the point you are making better of just remove the sentence.

➔ We deleted the last sentence.

Line 176: The last sentence is confusing. Please explain. Do you provide CALIPSO 532nm simulated signals as well as the 355 ATLID HSRL signals ? Is the depolarization ratio assumed to be the same for both wavelengths ?

➔ We provided the CALIPSO 532 nm simulated signals and the depolarization ratio for the only 532 nm signals. We modified the sentence more clearly.

Line 187: Delete "possible" there will certainly be noise in the real data !

➔ We deleted "possible".

Line 194: "could" ==> "can"

➔ We modified the sentence based on your comment.

Line 201: "The MSI..."

➔ We modified the sentence based on your comment.

Line 207: Six streams seems low for radiance simulations. Please comment on why this was done and the expected accuracy of the results.

➔ Our model (Rstar) implemented an accurate fast radiance correction method like Truncated Multiple and Single scattering approximation (TMS) method (Nakajima and Tanaka 1988), and can have errors less than 1 % with a very small stream number like six streams.

Line 220: "The MSI..."

➔ We modified the sentence based on your comment.

Line 224--229: Did you also consider the effect of the "smile" on the surface reflectance properties ?

➔ No, not in this study. We try to consider the smile effect on aerosol retrievals with surface reflectance over land in the other study.

Line 253: "The BBR has two channels; covering 0.25 to 50 um for ......and another covering from 0.25 to 4 um ....." Also. Replace "observing" by "estimating". Fluxes can not be physically observed. Only radiances can be observed and they are then used to estimate the TOA fluxes.

Also, please make it clear that radiances are not calculated.

➔ We modified the sentence based on your comment.

Section 4

=========

Line 280: "We have introduced the JAXA simulated EarthCARE L1 data that are ....."

➔ We modified the sentence based on your comment.

Lines 283--287. This is very unclear. Please re-write. Also, appropriate references should be added here. Perhaps Schutgens 2008 and Kollios 2018 ?

➔ We re-wrote "The EarthCARE CPR has a larger sampling volume than the ground observation and a fast motion. The inhomogeneous distribution of hydrometeors within the instantaneous field of view (IFOV) caused significant Doppler velocity biases (Schutgens 2007; Kollias et al., 2018). The effect of the inhomogeneous distribution in the IFOV on the Doppler velocity is different in along the track direction, and is referred to as the non-uniform beam filling effects (NUBFs). The impact of NUBFs on the Doppler velocity accuracy should be investigated for its improvement.". based on your comments.

Line 313: Delete ", which has limitation in representing scattering properties in 3D cloud fields". It is not the scattering properties you likely mean here, but rather the 3D radiation field.

➔ We modified the sentence based on your comment.

Summary

=======

Line 328: "The EarthCARE CPR...."

➡ We modified the sentence based on your comment.

Line 342: "The BBR..."

➡ We modified the sentence based on your comment.

Woosub Roh
Atmosphere and Ocean Research Institute, The University of Tokyo
5-1-5, Kashiwanoha, Kashiwa-shi, Chiba, Japan
Phone No: 81- 04-7136-4371
Fax No: 81- 04-7136-4375
Email Address: ws-roh@aori.u-tokyo.ac.jp

---

## Author Response (AR2)

To Reviewer:

We appreciate Reviewer's constructive comments.

**Public justification (visible to the public if the article is accepted and published**):

Thank you for your revised version of the paper. The figures are better but still need work to bring them to publication quality (see also Reviewer 1's second report) - this currently detracts from the paper. I'm not sure how you are producing these figures, but if your graphics program has the capability to output vector graphics (EPS or PDF) then please use this as it will avoid pixelation. Specific changes required:

1. Figure 1: Please remove the purple line: it makes the interpretation of the colour of the other line unclear, and adds no information to the figure since it is always next to the other line.

   ➜ We removed the purple line.

2. Please add units to all figures where requested by Reviewer 1.

   ➜ We added units in the captions of all figures.

3. Table 1: "clutter" and "noise" are uncountable nouns in this context, so should be singular not plural.

➜ We modified "clutter" and "noise" based on your comment.

4. Figure 3: The border of the panels is not fully visible: try printing with vector graphics. Titles of panels, if they are really needed, should not contain underscores or redundant information. If you must have panel titles then "CPR reflectivity (dBZ)" and "CPR Doppler velocity (m/s)" would be more appropriate. Finally, can you use a more appropriate aspect

ratio for panels b and c? The information is very squashed with these square panels. At least set the x range of the panel to cover only the range of the data, to avoid whitespace on the left and right.

→ We removed the titles of panels and made more clear borders.

5. Figure 5: Border of panels not fully visible. What does "dBZ_CON" mean? Please use meaningful panel titles if you must have panel titles.

→ We removed the titles of panels and made more clear borders.

6. Figures 6 and 7: Please use clear panel titles: no underscores, no unnecessary abbreviations, no redundant information that is common to all panels (don't say this Frame A because they all are). Fig. 7: Set the x range to cover only the range of the data to avoid whitespace on left and right.

→ We removed the titles of panels and the changed the x range.

7. Figures 7 and 8: You say these are log scales but are you plotting the natural logarithm or the base-10 logarithm? Please state in the captions.

→ We added "a base-10" in the captions in Figs. 7 and 8.

8. Figure 8: Panel border not fully visible. Panel titles not really needed as the information is in the caption and the difference is obvious. (Also note that "backscatter" and "noise" are both uncountable nouns in this context so should be singular.)

→ We removed the titles of panels and made more clear borders.

9. Figure 10: It looks like this has been stored as a JPEG with lossy compression: the text is blurred. Please save as a vector format with clean text.

→ We convert the figure as a vector format (emf).

10. Figure 11: Fix panel titles. Caption: "lie" -> "line".

→ We removed the panel titles and modified the caption.

11. Figure 13c,d: Fix titles to panels, change x scale to fit to data avoiding whitespace.

→ We removed the titles of panels and the changed the x range.

12. Data availability statement should probably reference the DOI like this "https://doi.org/10.5281/zenodo.7835229 (Roh et al. 2023)" and then add the reference in the references section (similar to how it is listed on the Zenodo page).

→ We modified the statement in Data availability and add the reference based on your comment.

13. I suggest you use "SMILE" in capitals throughout: at first use it is apparently an acronym.

→ We changed "SMILE" from "smile".

Woosub Roh
Atmosphere and Ocean Research Institute, The University of Tokyo
5-1-5, Kashiwanoha, Kashiwa-shi, Chiba, Japan
Phone No: 81- 04-7136-4371
Fax No: 81- 04-7136-4375
Email Address: ws-roh@aori.u-tokyo.ac.jp